# Radiation-Induced Nephropathy in the Murine Model Is Ameliorated by Targeting Heparanase

**DOI:** 10.3390/biomedicines11030710

**Published:** 2023-02-27

**Authors:** Alexia Abecassis, Esther Hermano, Kim Sheva, Ariel M. Rubinstein, Michael Elkin, Amichay Meirovitz

**Affiliations:** 1Sharett Institute of Oncology, Hadassah-Hebrew University Medical Center, Jerusalem 91120, Israel; 2Legacy Heritage Oncology Center and Dr. Larry Norton Institute, Soroka University Medical Center, Be’er Sheva 84101, Israel; 3Hebrew University Medical School, Jerusalem 91120, Israel

**Keywords:** heparanase, radiotherapy of cancer, pelvic malignancies, extracellular matrix, radiation nephropathy

## Abstract

Agents used to reduce adverse effects common in cancer treatment modalities do not typically possess tumor-suppressing properties. We report that heparanase, an extracellular matrix-degrading enzyme, is a promising candidate for preventing radiation nephropathy. Heparanase promotes tumor development and progression and is upregulated in tumors found in the abdominal/pelvic cavity, whose radiation treatment may result in radiation nephropathy. Additionally, heparan sulfate degradation by heparanase has been linked to glomerular and tubular/interstitial injury in several kidney disorders. In this study, heparanase mRNA levels were measured in HK-2- and HEK-293-irradiated kidney cells and in a murine radiation nephropathy model by qRT-PCR. Roneparstat (specific heparanase inhibitor) was administered to irradiated mice, and 24 h urinary albumin was measured. Kidneys were harvested and weighed 30 weeks post-irradiation. Clinically relevant doses of ionizing radiation upregulated heparanase expression in both renal cells and mice kidneys. A murine model of abdominal radiation therapy revealed that Roneparstat abolished radiation-induced albuminuria—the hallmark of radiation nephropathy. Given the well-documented anti-cancer effects of heparanase inhibition, our findings attest this enzyme to be a unique target in cancer therapy due to its dual action. Targeting heparanase exerts not only direct anti-tumor effects but protects against radiation-induced kidney damage—the backbone of cancer therapy across a range of malignancies.

## 1. Introduction

Radiation nephropathy (RN) is a significant side effect of radiation therapy when used in the treatment of pelvic malignancies such as gastrointestinal cancers, gynecologic cancers, lymphomas, sarcomas of the upper abdomen and total body irradiation [1]. The underlying mechanisms of RN pathogenesis as well as the mediators responsible for the deterioration of kidney function have not been fully elucidated. The notion that RN is mediated solely by DNA damage-related cell loss at division, and therefore is potentially unavoidable, has been transformed. This is due to the recognition that radiation-induced injury also involves complex and dynamic interactions between various cellular components (i.e., glomerular, tubular and interstitial) as well as the extracellular matrix (ECM) of renal tissue [2,3].

Heparanase is the sole mammalian endoglycosidase capable of degrading heparan sulfate (HS)—the principal polysaccharide of the ECM and cell surfaces in a wide range of tissues. HS chains play an important role in ECM integrity, barrier function and cell–ECM interactions, providing a structural framework for proper tissue organization and architecture. The heparanase-mediated cleavage of HS is best studied in the context of malignant tumor progression, where the enzyme has been shown to promote tumor growth and therapy resistance through multiple mechanisms. Heparanase overexpression (driven in human cancers by numerous molecular pathways [4,5,6,7,8]) is closely associated with enhanced aggressiveness and a poorer prognosis in several types of tumors, notably gastric [9]; colon [4]; ovarian [5] and cervical [6] carcinoma and retroperitoneal sarcoma [10]. These are precisely the tumor types where radiotherapy may lead to kidney damage and RN. These findings have highlighted the potential of heparanase to be a promising drug target, and heparanase-inhibiting compounds are currently being evaluated in clinical trials as anti-cancer drugs [11,12].

More recent studies have highlighted the pathogenic role of heparanase-mediated HS cleavage in renal disorders. In the kidney, HS contributes to the integrity and barrier functions of the basement membrane and glycocalyx, regulation of inflammatory responses and control of the availability of HS-binding chemokines, cytokines and growth factors sequestered in the ECM [13,14]. The degradation of HS by heparanase therefore has a significant effect on the development and progression of numerous kidney pathologies associated with both glomerular and tubular/interstitial injury [14,15,16]. The pathogenic action of heparanase involves damaging the glomerular filtration barrier function, fostering inflammation-mediated renal injury and promoting vessel destabilization and tubulo-interstitial fibrosis [14,15,16,17]. The induction of heparanase expression and enzymatic activity has been demonstrated in animal models of glomerulonephritis (i.e., puromycin amino–nucleoside-induced nephrosis and passive Heymann nephritis), adriamycin nephropathy, anti-glomerular basement membrane nephritis, diabetic nephropathy and acute kidney injury, as well as in patients with diabetic nephropathy, IgA nephropathy, minimal change disease, C3 nephropathy, lupus nephritis, membranous glomerulopathy, nondiabetic nephrotic syndrome and chronic kidney diseases, and kidney-transplanted patients. Moreover, heparanase deficiency eliminated the development of albuminuria and renal damage in mouse models of diabetic nephropathy and glomerulonephritis, while the neutralization of enzyme activity by specific inhibitors resulted in reduced proteinuria in animal models of diabetic and non-diabetic proteinuric kidney diseases [14,15,16]. Based on mounting evidence implicating heparanase in renal dysfunction, along with observations of clinically relevant doses of ionizing radiation (IR) inducing heparanase expression in certain cell types [18], we hypothesized that heparanase mediates the kidney-damaging effect of IR and could therefore serve as a potential therapeutic target for RN.

## 2. Materials and Methods

In vitro irradiation: HK-2 human proximal tubule epithelial cells [17] and HEK-293 human embryonic kidney cells (ATCC, Manassas, VA, USA) were routinely maintained in DMEM supplemented with 1 mM glutamine, 50 µg/mL streptomycin, 50 U/mL penicillin and 10% fetal calf serum (FCS) at 37 °C and 7.5% CO_2_. Prior to irradiation, cells were maintained for 16 h in serum-free medium and then irradiated using a 60Co Picker unit irradiator (1.56 Gy/min).

In vivo irradiation and murine radiation nephropathy model: Based on previous murine RN models [19,20,21], eleven-week-old female C3H/HeNHsd mice were housed under SPF conditions and received regular chow and water ad libitum. A dose of 10 Gy radiation (previously reported to induce radiation nephropathy in murine models [22]) was delivered to the anaesthetized mice via a brachytherapy afterloader (I192 Nucletron microSelectron HDR, Veenendaal, The Netherlands) using a bronchial sleeve applicator. On the bronchial sleeve, 1 cm dwell points were marked 1 cm apart with a 10 cm distance between each set of markers. Up to 5 sets were placed on each sleeve. These markers corresponded to the location of the kidneys inside the mice. The dose was calculated based on a 1.0 cm isodose line, with a 0.5 cm width silicon bolus placed above and below the sleeve to ensure dose homogeneity. The treatment field was designed to cover a specific banded area across the abdomen that included both kidneys, while shielding the rest of the body. The prescribed radiation dose was confirmed by film dosimetry. Variation in the dose within the kidneys was estimated to be within ±10% of the prescribed dose.

Subcutaneous injections of Roneparstat (kindly provided by Alessandro Noseda, Leadiant Biosciences S.p.A, Rome, Italy) were administered to mice in the experimental group (300 µg in 100 µL saline/mouse/injection, twice a day). Mice in the control group were injected with saline alone. For urine collection at indicated time points, mice were placed in metabolic cages for 24 h. Urinary albumin was measured using an ELISA kit (Bethyl Laboratories Inc, Montgomery, TX, USA).

Reverse transcription and quantitative RT-PCR (qRT-PCR): RNA isolation from both the cultured cells and the snap-frozen kidney tissue samples, and qRT-PCR, were performed as previously described (18). The following primers were used:

Human heparanase: Sense 5′-GTTCTAATGCTCAGTTGCTCCT-3′,

Antisense 5′-ACTGCGACCCATTGATGAAA-3′;

Mouse heparanase: Sense 5′-GGAGCAAACTCCGAGTGTATC-3′,

Antisense 5′-CAGAATTTGACCGTTCAGTTGG-3′; and

Human Egr1: Sense 5′-GAGCAGCCCTACGAGC-3′

Antisense 5′-AGCGGCCAGTATAGGT-3′.

Ethical approval: All animal experiments were approved by and performed in accordance with the Hebrew University of Jerusalem’s Institutional Animal Care and Use Committee.

## 3. Results

IR induces the expression of heparanase in cells of kidney origin in vivo. HK-2 and HEK-293 cells either remained untreated or were treated with clinically relevant doses of IR, after which heparanase mRNA levels were determined by qRT-PCR. As shown in Figure 1, a significant increase in heparanase expression was detected following cell exposure to IR.

The early growth response (Egr1) transcription factor has been previously shown to upregulate the expression of the heparanase gene by binding specifically to its regulatory region [23]. Additionally, IR has been reported to induce Egr1 in tumor-derived cells [24,25]. Interestingly, using qRT-PCR, we detected that IR upregulates Egr-1 levels in both HK-2 and HEK-293 cells (Figure 1C,D), suggesting that an Egr-1-dependent mechanism is responsible for radiation-induced heparanase expression in kidney cells.

IR upregulates renal heparanase expression in vivo. The above findings prompted an examination of the effect of IR on renal heparanase expression in vivo. For this purpose, experimental mice either remained untreated or were treated with bilateral kidney irradiation, as described in the methods section. Forty-eight hours post irradiation, the mice were sacrificed, their kidneys were excised, and heparanase expression in the renal cortex was assessed by qRT-PCR. As can be seen in Figure 2, a significant increase in heparanase expression was readily detected in the renal cortex of irradiated mice as compared to age-matched, non-irradiated, control mice. This confirms the ability of radiation to induce renal heparanase expression.

The inhibition of heparanase abolishes radiation-induced albuminuria in a murine model of RN. Next, we investigated the effect of the specific heparanase inhibitor Roneparstat (SST0001) on the development of proteinuria in irradiated mice. To investigate the effect of heparanase inhibition on RN, we utilized a well-characterized C3H/HeNHsd mouse model [22]. RN was induced using bilateral kidney irradiation (10 Gy) for a relatively conformal radiation dose with minimal exposure and damage to the bowel, which is crucial for long-term survival of the mice as well as clinical relevance of the experiment. Age-matched, non-irradiated mice were used as a control. Irradiated mice were treated with either Roneparstat or the vehicle control (saline) and 24 h albumin excretion was assessed at weeks 10, 20 and 30 of the experiment. As shown in Figure 3, at week 20, a marked and statistically significant increase in 24 h albumin excretion was noted in saline-treated irradiated vs. non-irradiated mice (*p* = 0.032). Interestingly, the administration of Roneparstat diminished this increase, where corresponding values of 24 h albumin excretion did not increase significantly in irradiated Roneparstat-treated mice as compared with the basal levels observed in non-irradiated mice (Figure 3). A significant difference in 24 h albumin excretion between saline-treated irradiated and non-irradiated control mice, but not between Roneparstat-treated and control mice, was maintained on week 30 of the experiment (Figure 3). Kidney irradiation also resulted in an absolute renal weight reduction of 10.5% in saline-treated mice as compared to IR-untreated mice at 30 weeks (although the differences between these groups did not reach statistical significance) (Figure 3).

## 4. Discussion

Radiation therapy forms one of the cornerstones of anticancer treatment modalities for a range of malignancies. Currently, more than 60% of cancer sufferers undergo radiation therapy either as a monotherapy or, more commonly, in combination with either chemotherapy or surgery [26,27]. Although ionizing radiation is highly effective in controlling tumor growth and prolonging overall survival, the exposure of healthy tissue to the radiation field results in unavoidable adverse effects. Despite advances in radiation delivery techniques, limiting ionizing radiation exposure to only cancerous tissues remains a major challenge [28]. Due to the proximity of the pelvic region to the kidneys, RT for the treatment of any pelvic malignancies carries the risk of inducing radiation nephropathy (RN). RN is a kidney injury caused by exposure to ionizing radiation that usually presents as chronic kidney disease a few months post-RT, which has the potential to evolve into end-stage renal disease. The damaging features of RN have been found histologically in the vascular, glomerular and tubulointerstitial regions of the kidney [29]. There is a critical need to not only improve RT delivery techniques to limit the exposure of healthy tissue to IR, but also to reveal new potential therapeutic targets for novel treatment options for RN.

The radiation-induced expression of heparanase has been found in cancerous cells [18] and it was, therefore, of interest in this study to investigate whether a similar mechanism exists in cells of kidney origin. A significant increase in heparanase expression was, in fact, found in vitro using HK-2 and HEK-293 kidney cells following clinically relevant doses of exposure to IR. Among several factors controlling heparanase expression, the early growth response (Egr1) transcription factor acts as an activator for the expression of the heparanase gene in several cell types, including kidney cells, where it binds to the heparanase promoter and activates heparanase expression [23]. Notably, Egr1 is rapidly induced in response to IR in several cancerous cell lines [24,25]. The present study confirmed this notion, whereby IR was found to upregulate Egr-1 levels in kidney cells, implicating an Egr-1-dependent mechanism in radiation-induced heparanase expression. These results were confirmed in vivo in mice using bilateral kidney irradiation, where a significant increase in heparanase expression was seen in the renal cortex.

The induction of heparanase expression by IR in kidney cells both in vitro and in vivo, together with the known contribution of heparanase to the pathogenesis of several kidney disorders other than RN [14,15,16], led us to hypothesize that the inhibition of heparanase may prevent the progression of RN. To validate this hypothesis, the specific heparanase inhibitor known as Roneparstat was administered to irradiated mice and the resultant effect on the development of proteinuria was assessed. Roneparstat, a 15–25 kDa N-acetylated and glycol split heparin, is one of the most potent and widely studied heparanase inhibitors that effectively inhibits heparanase enzymatic activity in vitro and is devoid of the anticoagulant activity of unmodified heparin. The effectiveness of Roneparstat in inhibiting the pathologic action of heparanase in vivo has been demonstrated in heparanase-driven processes other than RN, including malignant tumor progression and numerous non-malignant conditions [30]. In this study, Roneparstat successfully reduced both radiation-induced 24 h albuminuria as well as kidney weight loss, showcasing not only its ability to slow RN progression, but also the prominent role of heparanase in this disease. These findings are in agreement with previous observations in mouse models of RN [31], as well as in clinical studies where a progressive decrease in kidney size was documented in patients that had undergone abdominal radiation therapy [32]. Importantly, the administration of Roneparstat diminished renal weight loss, further implicating heparanase in RN and validating the inhibition of this enzyme as a promising approach to mitigate renal radiation injury. A limitation of the present study is that possible differences in RN occurrence and response to heparanase inhibition based on sex were not addressed in the in vivo model.

It should be noted that the heparanase enzyme was previously linked to the development and progression of essentially all tumor types found in the abdominal/pelvic cavity whose radiation treatment may lead to RN (i.e., gastric [9]; colon [4]; ovarian [5] and cervical [6] carcinoma, pancreatic cancer [7,33], retroperitoneal sarcoma [10] and hepatobiliary tumors [34]). Moreover, in these tumor types, inhibitors of heparanase have exerted anti-cancer effects in preclinical models and are currently being tested clinically [34,35,36]. In the setting of the above-mentioned cancer types, our findings highlight heparanase as a unique target among the extracellular matrix molecules. Due to the dual action of heparanase, the inhibition of this enzyme, when administered concomitantly with radiation therapy, is expected to exert not only direct anti-tumor effects [8] but also to protect against kidney damage induced by radiation—the backbone of cancer therapy across a broad range of abdominal/pelvic malignancies. Further studies are warranted to validate this heparanase-based therapeutic approach and to optimally investigate its potential.

## Figures and Tables

**Figure 1 biomedicines-11-00710-f001:**
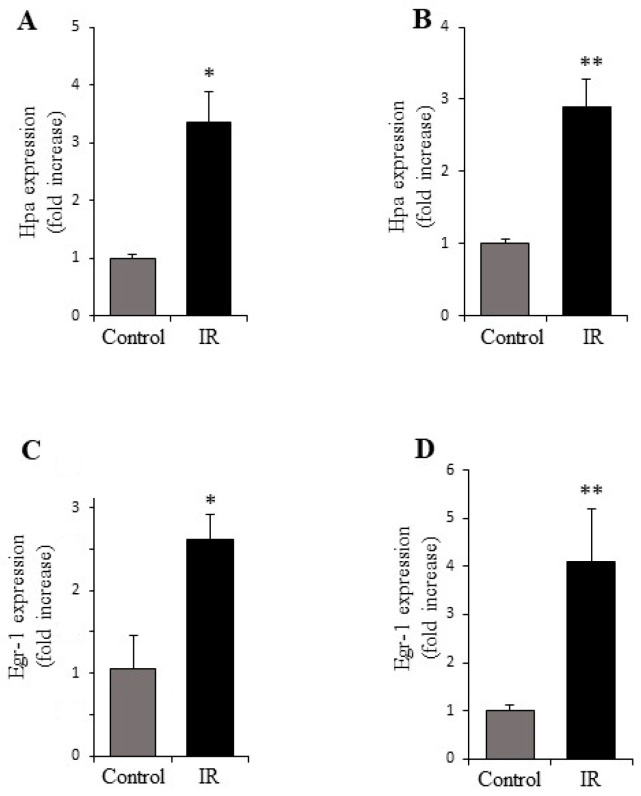
IR induces heparanase expression in kidney-derived cell lines. Prior to IR treatment, HK-2 human proximal tubule epithelial cells (**A**,**C**) and HEK-293 human embryonic kidney cells (**B**,**D**) were maintained for 16 h in serum-free medium. Cells either remained untreated (control) or were irradiated (IR) with 5 Gy (**A**,**C**) and 10 Gy (**B**,**D**). Heparanase (Hpa) and Egr-1expression were assessed by qRT-PCR. Error bars represent ± SE; * *p* ≤ 0.003, ** *p* = 0.013.

**Figure 2 biomedicines-11-00710-f002:**
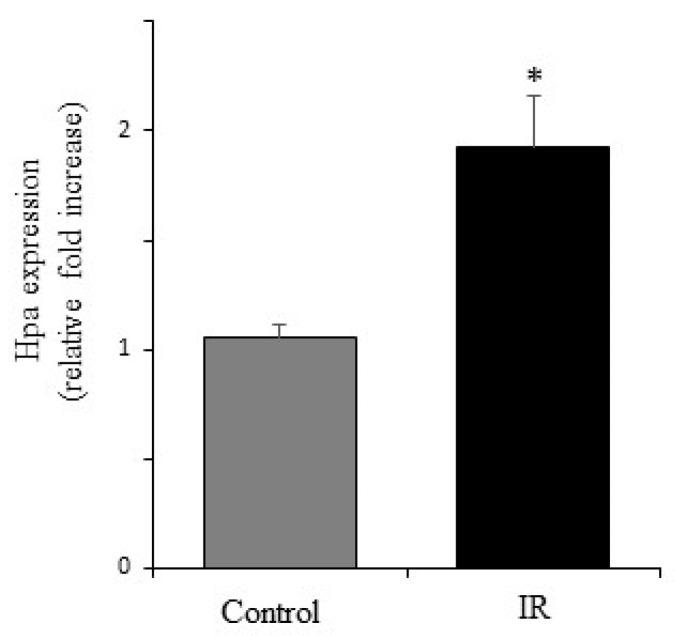
IR induces heparanase expression in the kidneys of irradiated mice. C3H/HeNHsd mice remained untreated (grey bar) or were treated with conformal bilateral kidney irradiation as described in Methods (black bar). Mice were sacrificed and the renal cortex from each kidney was extracted and stored at −80 °C. Heparanase expression was assessed by qRT-PCR. Error bars represent ± SE; n = 3 mice per group; * *p* < 0.008.

**Figure 3 biomedicines-11-00710-f003:**
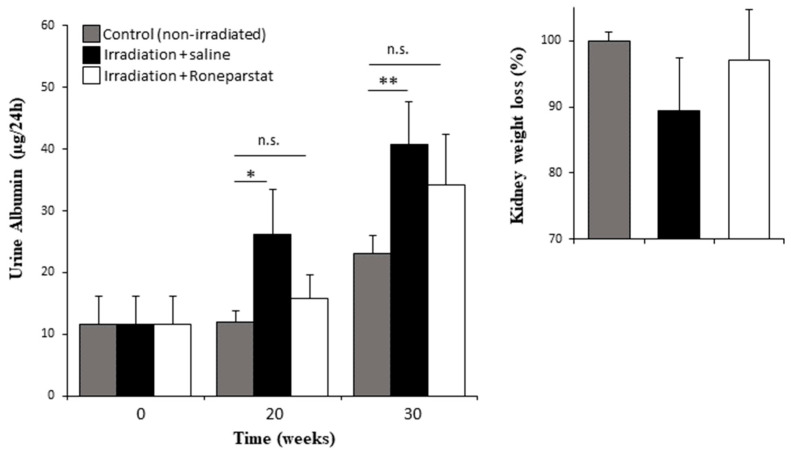
Effect of the heparanase inhibitor Roneparstat on urinary albumin excretion in the radiation nephropathy murine model. C3H/HeNHsd mice remained untreated (grey bars) or were treated with conformal bilateral kidney irradiation (as described in Methods) in combination with either the vehicle control (saline) alone (black bars) or with Roneparstat (white bars), administered daily for 30 weeks. Prior to irradiation (week 0) and at 20 and 30 weeks post-irradiation, 24 h urine samples were collected, and urinary albumin content was measured. Error bars represent ± SE; n ≥ 5 mice per group; * *p* = 0.03, ** *p* = 0.018, n.s.—not statistically significant. Inset. Roneparstat diminishes IR-induced renal weight reduction. Kidneys from untreated mice (grey bars) or those irradiated and treated with either the vehicle control (saline) alone (black bars) or Roneparstat (white bars) were harvested and weighed 30 weeks post-irradiation. Data represent the mean left kidney weight ± SE.

## Data Availability

Research data are stored in an institutional repository and will be shared upon request to the corresponding author.

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
