# Peer review of "Radiation-Induced Nephropathy in the Murine Model Is Ameliorated by Targeting Heparanase"

_biomedicines, 2023, doi:10.3390/biomedicines11030710_

Round 1

Reviewer 1 Report

Major Comments:

1.      Why did the authors choose female mice? Is there any specific reason? The authors should do experiments on male mice also to avoid potential sex biased results.

2.      As IR nephropathy is common in cancer patients why didn’t the authors develop a cancer model first followed by IR for their study? This could have better mimicked the real case.

3.      Why heparanase is increased in the tumors of the pelvic or abdominal cavity?

4.      Did the authors measure other ECM degrading enzymes and ECM proteins during IR nephropathy. If not, please do necessary experiments and provide those data.

5.      Authors may discuss about the heparanase levels in IR nephropathy with other types of nephropathies.

6.      As protein (enzyme) is the functional unit of the cellular processes and mRNA expression levels may not be completely converted into the protein levels, the authors should provide protein expression levels of the mRNA data. This is very important.

Minor Comments:

1.      Please check the manuscript for grammatical errors.

Author Response

Comments and Suggestions for Authors

Major Comments: 

  1. Why did the authors choose female mice? Is there any specific reason? The authors should do experiments on male mice also to avoid potential sex biased results.

OUR RESPONSE: The reason we used female mice was because in all previous publications describing the RN murine model utilized in our study only female mice were utilized. As we were using an existing in vivo model we followed suit. The aforementioned publications have additionally been added as references in the ‘Materials and Methods’ section of our manuscript (New references 19 -21).

In addition, there are studies that have demonstrated that no differences in renal structural and functional damage could be detected between male and female mice in other types of nephropathies (e.g., https://pubmed.ncbi.nlm.nih.gov/30810354/).

Furthermore, to repeat the study in male mice as suggested will take a significant amount of time (20-30 weeks are needed to fully evaluate RN and effects of heparanase inhibition). Our concern is that the requested in vivo experiment is beyond our capabilities and timeframe. Given the above considerations,  we ask the reviewer to agree that we add the following statement to the Discussion section: "The limitation of the present study is that possible differences in RN occurrence and response to heparanase inhibition based on sex were not addressed in the in vivo model" (lines 250-252).

  1. As IR nephropathy is common in cancer patients why didn’t the authors develop a cancer model first followed by IR for their study? This could have better mimicked the real case.

OUR RESPONSE: We agree that first establishing a cancer model which is then followed by IR treatment would better mimic a real case. Unfortunately, it is essentially impossible to establish xenograft mouse model in which animals will survive 20-30 weeks after tumor establishment (the time necessary to detect RN, as in our study).

  1. Why heparanase is increased in the tumors of the pelvic or abdominal cavity?

OUR RESPONSE: As in the cases of many other tumor-promoting/tumor associated genes, increased expression of heparanase in various tumor types (including the tumors of the pelvic / abdominal cavity) is driven by multiple mechanisms. In the revised version of the manuscript, we clearly indicate this fact and provide references for a more detailed description of the relevant mechanisms.

  1. Did the authors measure other ECM degrading enzymes and ECM proteins during IR nephropathy. If not, please do necessary experiments and provide those data.

OUR RESPONSE: Other ECM-degrading enzymes have multiple substrates and isoenzymes. HS is the main component of the ECM and has only one known enzyme that degrades it, and that is the reason for focusing solely on heparanase in this study.  This approach is widely considered sufficient to provide initial proof-of-concept in basic studies, especially when backed by in vivo data.

  1. Authors may discuss about the heparanase levels in IR nephropathy with other types of nephropathies.

OUR RESPONSE: Heparanase levels/involvement in other types of nephropathies is discussed in lines 66-73 of the revised manuscript.             

  1. As protein (enzyme) is the functional unit of the cellular processes and mRNA expression levels may not be completely converted into the protein levels, the authors should provide protein expression levels of the mRNA data. This is very important.

OUR RESPONSE: We have previously shown that there is a direct correlation between heparanase enzyme levels to mRNA levels in renal cells (please see: PMID: 22106160). Additionally, effects of the enzymatic inhibitor of heparanase  (i.e., Roneparstat) reported in the present study support the notion that our mRNA expression data  properly reflects an increase in levels of enzymatically-functional heparanase protein. Therefore, we feel that additional protein analysis is not necessary in this specific case and hope the Reviewer will accept this reasoning.

Minor Comments:

  1. Please check the manuscript for grammatical errors.

OUR RESPONSE: The manuscript has been checked and edits have been made. Please accept our apology for not being careful enough when the manuscript was first submitted.

Reviewer 2 Report

This study investigated the mechanism underlining of radiation induced nephoropathy in a murine model. The rational behind the experiment was clear and straight forward. The manuscript is almost well written. 

The authors should mentioned in the method section of the abstract more details about the methods employed. 

There are some minor grammar issues that should be fixed in order to aid the accessibility of the results to

the reader.

Please describe better the In-vivo irradiation and murine radiation nephropathy model paragraph

Author Response

Comments and Suggestions for Authors

  1. This study investigated the mechanism underlining of radiation induced nephoropathy in a murine model. The rational behind the experiment was clear and straight forward. The manuscript is almost well written. 
  2. The authors should mentioned in the method section of the abstract more details about the methods employed.

OUR RESPONSE: Further information was added to the abstract.  

  1. There are some minor grammar issues that should be fixed in order to aid the accessibility of the results to the reader.

OUR RESPONSE: The manuscript has been checked and language edits have been made.

  1. Please describe better the In-vivo irradiation and murine radiation nephropathy model paragraph

OUR RESPONSE: Additional information regarding the administered radiation has been added. Please see lines 95-97 of the revised manuscript.

 Additionally, as requested, the text of the manuscript was carefully checked/corrected by English speaking editing professional.

Reviewer 3 Report

Thank you to Abecassis for submitting this manuscript examining roneparstat as a potential agent to minimise radiation-induced nephropathy. I have some queries:

1) I strongly suggest separating the Results from a distinct and separate Discussion section

2) Figure 3 - please add "urine albumin" to the y axis

3) Figure 3 - was the difference between the irradiation/saline group vs the irradiation/roneparstat statistically significantly different at 20 or 30wks? This is required to be able to in any way assert that roneparstat has an effect.

4) Please indicate if the kidney weight of

 - control vs irradiation/saline group is significant or not

 - control vs irradiation/roneparstat group is significant or not

- irradiation/saline group  vs irradiation/roneparstat group is significant or not

5) Was there any measure of kidney function such as creatinine or urea amongst the groups? 

6) Was there any histological comparison amongst the groups?

7) How many mice were in each of the study groups?

Author Response

1) I strongly suggest separating the Results from a distinct and separate Discussion section

OUR RESPONSE: As requested, the Results and Discussion are now separate.

2) Figure 3 - please add "urine albumin" to the y axis

OUR RESPONSE: We apologize for this oversight - the y axis has been edited accordingly.

3) Figure 3 - was the difference between the irradiation/saline group vs the irradiation/roneparstat statistically significantly different at 20 or 30wks? This is required to be able to in any way assert that roneparstat has an effect.

OUR RESPONSE: The differences between irradiated/saline-treated vs the irradiated/roneparstat-treated mice did not reach statistical significance. As stated in lines 173-181, the conclusion regarding Roneparstat effect was drawn based on the lack of a statistically significant difference between irradiated Roneparstat-treated vs control (non-irradiated) mice, in the face of the statistically significant difference between irradiated (saline-treated) vs. control mice. This approach is often considered sufficient to provide initial proof-of-concept in basic studies involving complex and long-lasting mouse models of human diseases.

4) Please indicate if the kidney weight of

 - control vs irradiation/saline group is significant or not

 - control vs irradiation/roneparstat group is significant or not

- irradiation/saline group vs irradiation/roneparstat group is significant or not

OUR RESPONSE: the differences between the above-mentioned groups did not reach statistical significance (lines 183-184 in the revised manuscript).

5) Was there any measure of kidney function such as creatinine or urea amongst the groups? 

OUR RESPONSE: In mice creatinine levels are not reliable enough for monitoring nephropathy. Albuminuria is a better surrogate marker for measuring nephropathy.

6) Was there any histological comparison amongst the groups?

OUR RESPONSE: No. In prior studies it was noted that the damage at this disease stage is not well reflected histologically, where more time and progression is required to see anatomical changes.

7) How many mice were in each of the study groups?

OUR RESPONSE: The number of mice per group (≥ 5) is indicated in line 194 of the revised manuscript.

Round 2

Reviewer 1 Report

The manuscript is now improved and can be accepted for publication.

Reviewer 3 Report

Thank you for responding to the revision queries. I am concerned that not all results are being reported, including those that are not statistically significant.